# Pathological Characteristics, Management, and Prognosis of Rectal Neuroendocrine Tumors: A Retrospective Study from a Tertiary Hospital

**DOI:** 10.3390/diagnostics14171881

**Published:** 2024-08-28

**Authors:** Federica Cavalcoli, Emanuele Rausa, Davide Ferrari, Roberto Rosa, Marco Maccauro, Sara Pusceddu, Giovanna Sabella, Paolo Cantù, Marco Vitellaro, Jorgelina Coppa, Vincenzo Mazzaferro

**Affiliations:** 1Gastroenterology and Gastrointestinal Endoscopy Unit, Fondazione IRCCS Istituto Nazionale dei Tumori, Via Venezian 1, 20133 Milan, Italy; federica.cavalcoli@istitutotumori.mi.it (F.C.); roberto.rosa@istitutotumori.mi.it (R.R.); paolo.cantu@istitutotumori.mi.it (P.C.); 2Unit of Hereditary Digestive Tract Tumors, Fondazione IRCCS Istituto Nazionale dei Tumori, Via Venezian 1, 20133 Milan, Italy; emanuele.rausa@istitutotumori.mi.it (E.R.); marco.vitellaro@istitutotumori.mi.it (M.V.); 3Colorectal Surgery Division, Fondazione IRCCS Istituto Nazionale dei Tumori, Via Venezian 1, 20133 Milan, Italy; 4Departement of Nuclear Medicine, Fondazione IRCCS Istituto Nazionale dei Tumori, Via Venezian 1, 20133 Milan, Italy; marco.maccauro@istitutotumori.mi.it; 5Department of Medical Oncology, Fondazione IRCCS Istituto Nazionale dei Tumori, Via Venezian 1, 20133 Milan, Italy; sara.pusceddu@istitutotumori.mi.it; 6Department of the Pathology and Laboratory Medicine, Fondazione IRCCS Istituto Nazionale dei Tumori, Via Venezian 1, 20133 Milan, Italy; giovanna.sabella@istitutotumori.mi.it; 7HPB Surgery, Hepatology and Liver Transplantation, Fondazione IRCCS Istituto Nazionale dei Tumori, Via Venezian 1, 20133 Milan, Italy; jorgelina.coppa@istitutotumori.mi.it (J.C.); vincenzo.mazzaferro@istitutotumori.mi.it (V.M.)

**Keywords:** neuroendocrine tumors, rectal cancer, neuroendocrine management, tumor size, tumor grade, overall survival, progression-free survival

## Abstract

Background: Rectal neuroendocrine tumors (rNENs) are rare, constituting 1–2% of rectal tumors, and are often asymptomatic, leading to challenges in early diagnosis. Current management guidelines recommend endoscopic resection for small lesions and surgical intervention for larger or high-risk tumors. This study aims to retrospectively analyze the pathological characteristics, management, and prognosis of rNEN patients. Methods: Data from the Neuroendocrine Tumor Registry at a tertiary hospital in Milan, Italy from 2005 to 2023 were retrospectively analyzed. Patient demographics, disease characteristics, pathology findings, treatment details, and surveillance data were collected. Statistical analyses included descriptive statistics, multivariable binary logistic regression, and Kaplan–Meier survival analysis. Results: Forty-five patients were included, 53.3% male with a mean age of 57.5 years. Most patients were asymptomatic, with incidental diagnosis during colonoscopy. Endoscopic excision was the primary treatment modality (77.8%), with surgical resection reserved for incomplete or inappropriate endoscopic resections. Disease progression occurred in 13 patients (28.9%), with tumor-related mortality of 22.2%. Kaplan–Meier analysis showed 5- and 10-year survival rates of 68.8% and 59.1%, respectively, with corresponding progression-free survival rates of 72.8% and 54.0%. Tumor stage was significantly associated with disease progression on multivariable analysis (OR = 7.230, *p* = 0.039). Conclusions: This study highlights the heterogeneous presentation and prognosis of rNENs, with a substantial proportion diagnosed incidentally. Endoscopic management was predominantly utilized, aligning with current guidelines for localized tumors. Tumor stage emerged as a significant predictor of disease progression, emphasizing the importance of accurate staging for optimal management. Further research is warranted to refine management protocols and validate these findings.

## 1. Introduction

Rectal neuroendocrine tumors (rNENs), constituting 1–2% of rectal tumors, significantly contribute to gastrointestinal NEN and are prevalent in individuals aged 40–60, typically presenting as solitary tumors without distinctive clinical symptoms [1]. The rNENs primarily manifest as lesions smaller than 1 cm localized to the submucosa. Metastatic occurrences are less than 20%, and risk factors include size, atypical appearance, grade, and depth of invasion [2]. While surgical intervention remains the primary choice, the evolving landscape of endoscopic technology has led to an increasing preference for endoscopic treatment due to its reduced invasiveness, quicker recovery, and cost-effectiveness [3]. In fact, tumor size guides the management of rNENs, with the European Neuroendocrine Tumor Society (ENETS) [4] recommending endoscopic resection (endoscopic mucosal resection [EMR], endoscopic submucosal dissection [ESD], and endoscopic full-thickness resection [eFTR]) for lesions measuring 1 cm or less. For tumors ranging from 1 to 2 cm, particularly those exhibiting a heightened risk of invasion, pre-procedural assessments including full-imaging and endoscopic ultrasound (EUS) are advised, and then, multidisciplinary evaluation should be undertaken [4,5,6]. Post-endoscopic resection surveillance for rNENs involves a comprehensive patient history, physical examination, and consideration of a multiphasic CT scan or MRI. Re-evaluation occurs 3 to 12 months post-resection, followed by subsequent assessments every 6 to 12 months for up to a decade [6]. On the other hand, rNENs exceeding 2 cm, infiltrating the muscularis propria, or with possible loco-regional metastatic lymph nodes, necessitate low anterior resection (LAR) or, in the case of infiltration of the anal sphincter, abdominoperineal resection (APR) [7]. This study aims to retrospectively analyze the pathological characteristics, management, and prognosis of rNEN patients at our institute.

## 2. Materials and Methods

This study followed the Strengthening the Reporting of Observational Studies in Epidemiology (STROBE) Statement guidelines for reporting observational studies. After the Institutional Review Board approval was achieved (INT 220/19), data from the prospectively maintained Neuroendocrine Tumor Registry at the Fondazione IRCCS Istituto Nazionale dei Tumori of Milan (Italy) were retrospectively accessed from 1 January 2005 to 31 December 2023, and only data regarding rNENs were queried. Written informed consent for research purposes was routinely obtained from patients diagnosed with neuroendocrine tumors at our institution. The following variables were collected: characteristics of patients [date of birth, gender, BMI, comorbidities, medications used, and smoking status], disease characteristics [date of diagnosis, symptoms at presentation, clinical stage, distance from anal verge, size, and basal chromogranin A], pathology findings [positive or negative margins, tumor grade, and ki-67], treatment characteristics [endoscopic or surgical resection, medical treatment, and Peptide Receptor Radionuclide Therapy (PRRT)], surveillance data [date of last follow-up, date and site of recurrence or progression, and mortality]. The normal upper limit for chromogranin was considered 95 ng/mL [8]. All the neoplasms were retrospectively classified on the basis of immuno-histochemical characteristics, according to the WHO 2019 classification, based on the Ki-67 index and mitotic count (G1: Ki67 < 3% or mitotic count < 2 in 2 mm^2^, G2: Ki67 3–20% or mitotic count 2–20 in 2 mm^2^, and G3: Ki67 > 20% or mitotic count >20 in 2 mm^2^) [9] and were staged according to the tumor–node–metastasis (TNM) classification [10].

### Statistical Analysis

Continuous variables were described as means (±standard deviation), while categorical variables were reported as frequencies and percentages. Analyses were performed to identify potential factors associated with disease progression using a multivariable binary logistic regression, which included patients with significant *p* values on univariate analysis, expressed as odds ratio (OR) and 95% confidence intervals (CI). *p* values < 0.05 were considered significant for all comparisons. The Kaplan–Meier method calculated overall and progression-free survival curves. For overall survival, an event was defined as patient death. For progression-free survival, an event was defined as local recurrence or de novo metastasis. 

## 3. Results

A total of 45 patients were identified and included in the final study population. Patient demographics and clinical presentation characteristics are summarized in Table 1, while the tumor management is described in Table 2.

### 3.1. Demographics and Baseline Characteristics

Among the 45 patients, the majority were male (53.3%, *n* = 24). Twenty-nine patients had comorbidities: cardiovascular (*n* = 15), previous tumors (*n* = 8), pulmonary (*n* = 6), diabetes (*n* = 3), ulcerative colitis (*n* = 2), or other comorbidities (*n* = 14). Overall, 11 patients (24.4%) presented for symptoms, which included abdominal pain (*n* = 5), rectal bleeding (*n* = 4), back pain, mass presence, and constipation (*n* = 1), while the remaining 34 patients were diagnosed incidentally on endoscopy. These colonoscopies were either performed for screening, for follow-up of colonic polyps, diverticulosis, ulcerative colitis, or previous colorectal cancer, and evaluation before hemorrhoidectomy. No patients presented with symptoms suggestive of hormonal secretion.

### 3.2. Diagnostic Work-Up

All diagnoses were performed through a colonoscopy, and the mean distance of the tumors from the anal verge was 7.0 ± 4.2 cm. Histology was obtained from all patients, and tumor grade was more frequently 1 (*n* = 27, 60%), followed by 3 (*n* = 12, 26.7%) with 11 patients having a poorly differentiated NEC and 1 having a well-differentiated NET. Baseline Chromogranin A was above normal levels in five patients (11.1%). All patients completed clinical staging with imaging, either with CT scan and MRI or EUS. The most frequent clinical stage was stage 1 (*n* = 28, 62.2%), followed by stage 4 (*n* = 11, 24.4%), stage 3 (*n* = 5, 11.1%), and stage 2 (*n* = 1, 2.2%). 

### 3.3. Individual Patient Management

Thirty-five patients (77.8%) underwent tumor excision, the majority endoscopically (*n* = 26, 57.8%), which included polypectomy (*n* = 15), endoscopic mucosal resection (*n* = 8), and endoscopic submucosal dissection (*n* = 3). 

In the subgroup of patients who underwent endoscopic treatment, median size was 5 mm (range 2–15), with all the patients being at stage I. Most of the patient had a G1 tumor (22, 92.3%) with one patient having a G2 (3.8%) and one a G3 tumor (3.8%), the latter being diagnosed after endoscopic resection. Radical endoscopic resection (R0) was obtained in 23/26 (88.5%) of cases. Of these, two underwent further surgical resection, while one patient remained in follow-up. Among the patients who had upfront surgery, five underwent TAMIS, and four underwent demolitive surgical resection: low anterior resection (*n* = 3) and proctocolectomy (*n* = 1). Among these patients, two had debulking surgery for the presence of distant metastases at the moment of surgery. Among the 10 patients (22.2%) who had nonoperative management, 8 underwent chemotherapy, 1 underwent peptide receptor radionuclide therapy, and 1 underwent somatostatin receptor antagonist therapy. Figure 1 and Figure 2 show an overview of treatments that patients underwent. 

### 3.4. Patient Outcomes and Survival Analysis 

With a mean follow-up of 46.6 ± 41.0 months, 13 patients experienced disease progression, which was in 2 cases represented by local recurrence, in 11 cases a de novo metastasis, and in 2 cases local recurrence was found with a synchronous de novo metastatic lesion. Tumor-related mortality was 22.2% (*n* = 10), and overall mortality was 24.4% (*n* = 11). Kaplan–Meier survival analysis (Figure 3) showed a 5- and 10-year survival rate of 68.8% and 59.1%, respectively, and a 72.8% and 54.0% 5- and 10-year progression-free survival rate (Figure 4).

### 3.5. Risk Factors for Disease Progression

Table 3 shows a multivariable binary logistic regression including all statistically significant factors associated with progression based on univariable analysis. Tumor grade, tumor stage, symptomatic presentation, tumor size, positive margins (R1 or R2), and Ki-67 were significant at univariable analysis and were included in the multivariable model, with the exclusion of positive margins due to the small number of observed cases. After multivariate analysis, only tumor stage was significantly associated with progression, with an OR = 7.230 for every classification change (*p* = 0.039).

## 4. Discussion

The present study reports data from a large series of rNEN patients with long-term follow-up in a Tertiary Care Center allowing for a better characterization of the complex clinical management in the real-life setting of these patients. Our findings underline the heterogeneity in terms of presentation and prognosis of rNENs. Consistent with previous studies, our study showed a slight male predominance and a median age for rNEN diagnosis between 50 and 60 years [11]. The vast majority of patients in this cohort were diagnosed incidentally during diagnostic colonoscopy, which is reflective of the often asymptomatic presentation of early-stage rNENs. This incidental detection underscores the importance of endoscopic screening and surveillance in early detection of potentially malignant lesions, particularly in individuals with risk factors such as previous colorectal cancer or chronic inflammatory conditions such as ulcerative colitis. Symptomatic presentation, most frequently abdominal pain and rectal bleeding, was observed in 24.4% of patients and was associated with a more advanced stage at diagnosis and a greater risk of progression at univariate analysis. This variability in presentation highlights the challenges in early clinical diagnosis of rNEN and suggests that a more aggressive management and timely initiation of systemic treatment would be advisable in patients with tumor-related symptoms. The management of rNEN in our series primarily involved endoscopic approaches, with a rate of R0 resection of 88.5%. Surgical interventions were reserved for cases where endoscopic resection was incomplete or inappropriate. This treatment strategy aligns with current guidelines that recommend a minimally invasive approach for small (<10/15 mm), low-grade, localized tumors to optimize outcomes and minimize recurrence risks [4]. Data from these series confirmed an excellent prognosis for endoscopically treated neoplasms, without any tumor-related death in this group and recurrence in a single case which presented with a higher grade at diagnosis (G3). The use of advanced endoscopic techniques, such as EMR and ESD, reflects the evolving landscape of endoscopic management which can, in selected cases, be used in sequence for unclear resection margins. Notably, the amplification of the endoscopic armamentarium allows for effective tumor control by reducing the necessity for surgical resections [12].

In consideration of higher grades, advanced stages, or the impossibility of achieving appropriately clear endoscopic resection margins, a surgical excision of the rectum with the loco-regional lymph nodes is mandatory [13]. In our series, LAR with total mesorectal excision was the most frequently used laparoscopic surgical approach.

While most rNENs are diagnosed in an early stage, as confirmed in our series, the treatment strategy is currently defined on the basis of the metastases-predicting factors, such as size of the neoplasm, endoscopic features, and lympho-vascular infiltration [14]. In rNENs where distant metastases are present, the primary approach typically involves systemic therapy, including somatostatin receptor antagonist, peptide receptor radionuclide therapy, chemotherapy, and everolimus/sunitinib [15,16]. The aim of systemic therapy in this scenario is to control the disease and alleviate symptoms [17,18]. Since rNENs are usually non-functional, systemic therapies primarily target their proliferative tendencies rather than symptom relief [19]. Traditionally, rNENs are reported as small tumors with indolent behavior in up to 90% of cases [20]. In our series, 35.5% of patients were diagnosed at advanced stages, possibly reflecting a selection bias for referral in an Oncologic Tertiary Care Center. Accordingly, Kaplan–Meier survival analysis indicated a 10-year survival rate of 59.1%, which is slightly lower compared to the 82.3% reported in the 2016 SEER analysis [21]. Survival analysis in our series is reflective of the generally favorable prognosis for well-differentiated early-stage NENs, but also points to the considerable risk of long-term mortality in more advanced cases. The relatively high rate of de novo metastasis among patients with disease progression underscores the aggressive nature of higher-grade NENs and the need for vigilant surveillance and possibly more aggressive therapeutic approaches in these cases. The prognosis of rNENs in our study was significantly influenced by tumor stage, with higher stages associated with an increased risk of progression and poorer survival outcomes. This finding is consistent with the existing literature, which highlights tumor stage as a critical determinant of survival in NEN patients [22]. The significant association of tumor stage with disease progression emphasizes the importance of accurate staging and the potential benefits of early detection [19,23]. Symptomatic presentation, tumor size, tumor grade, and Ki-67 were significant at univariable analysis but did not retain significance at multivariate analysis, and their impacts need to be evaluated in larger studies with longer follow-up.

This study has some limitations to consider in order to interpret the results. First, this study has a retrospective design and covers an extended period of time; this means that different pathological and clinical classification have been updated during the study period, and this may contribute to some heterogeneity in the patient’s management. Furthermore, new treatments, including endoscopic techniques and systemic treatment, have been developed in the last years. Finally, the small sample size is a major limitation. On the other hand, our study provides a faithful image of the evolving scenario of rNEN managements in the last 20 years. In addition, our Institute is a Tertiary Care Oncologic Center, and for this reason, it is possible that advanced patients may be referred for second opinions leading to a higher rate of metastatic/advanced rectal NEN in the present series.

## 5. Conclusions

In conclusion, our study highlights the complexity of diagnosing and managing rNENs, underscoring the critical role of early detection and appropriate staging in optimizing patient outcomes. The relatively high rate of advanced rectal NENs observed in our series suggests that rNENs may have a less indolent behavior than usually reported, and appropriate risk stratification is necessary to plan a correct treatment strategy. Further studies are required to validate our findings and potentially refine management protocols, particularly concerning the role of novel systemic treatments in advanced cases.

## Figures and Tables

**Figure 1 diagnostics-14-01881-f001:**
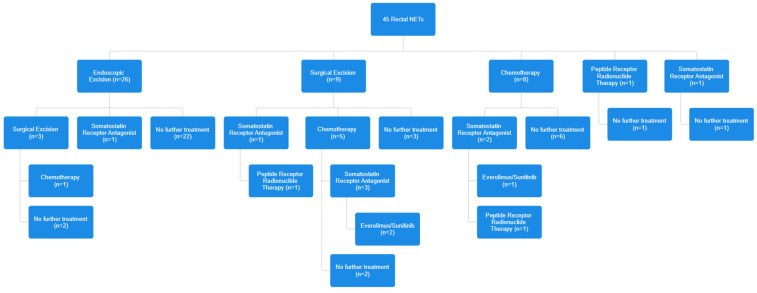
Overview of treatment for individual tumors. No further treatment indicates complete response or palliation.

**Figure 2 diagnostics-14-01881-f002:**
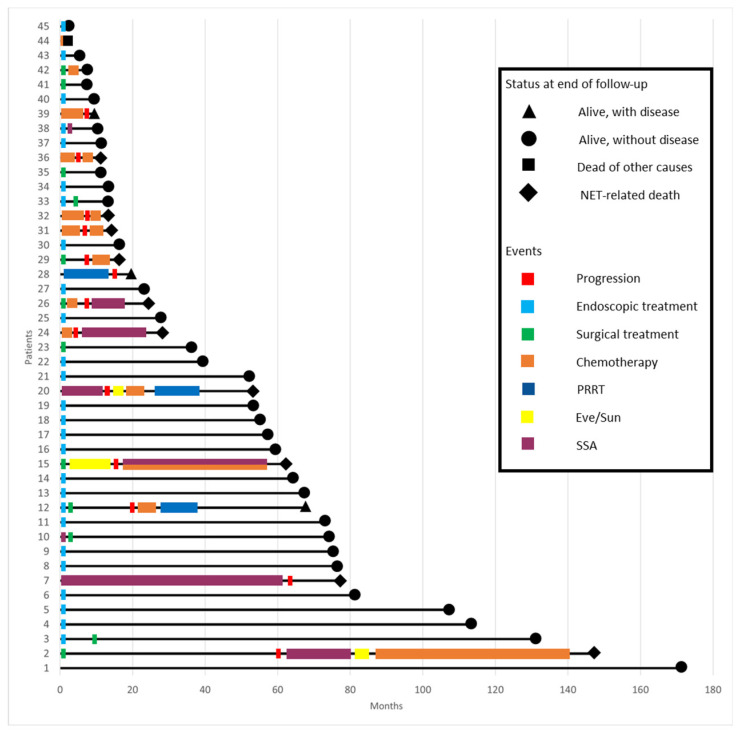
Swimmer plot representing the treatment courses of individual patients. Individual treatments are denoted as bars, while the length of the bars represent the length of treatment. The black lines represent the periods of active observation. PRRT: peptide receptor radionuclide therapy; SSA: somatostatin receptor antagonist, and Eve/Sun: everolimus/sunitinib.

**Figure 3 diagnostics-14-01881-f003:**
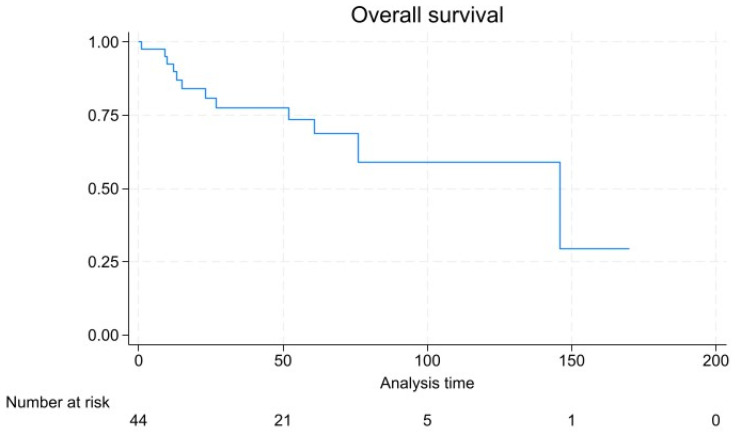
Kaplan–Meier plot for overall survival.

**Figure 4 diagnostics-14-01881-f004:**
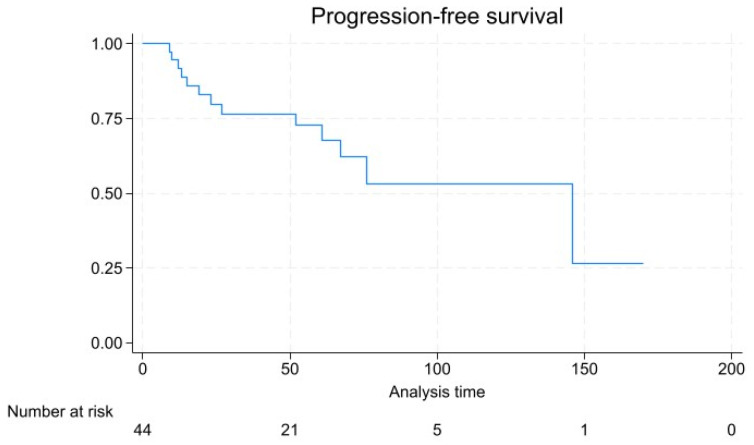
Kaplan–Meier plot for progression free survival.

**Table 1 diagnostics-14-01881-t001:** Patient demographics, presentation, and intervention.

Demographics	*n* = 45
Male	24 (53.3)
Female	21 (46.7)
Age at diagnosis, mean ± SD (years)	57.5 ± 13.5
BMI, mean ± SD (kg/m^2^)	24.6 ± 5.3
Smoker	7 (15.6)
**Presentation**	
Incidental finding	34 (75.6)
Symptomatic	11 (24.4)
Rectal bleeding	4 (8.9)
Abdominal pain	5 (11.1)
Back pain	1 (2.2)
Mass presence	1 (2.2)
Constipation	1 (2.2)
Distance from anal verge, mean ± SD (cm)	7.0 ± 4.2

Values are *n* (%) unless otherwise indicated.

**Table 2 diagnostics-14-01881-t002:** Outcomes.

Intervention	*n* = 45
Endoscopic resection	26 (57.8)
Polypectomy	15 (57.7)
EMR	8 (30.8)
ESD	3 (11.5)
Surgical resection	9 (20)
TAMIS	5 (55.6)
LAR	3 (33.3)
Proctocolectomy	1 (11.1)
No intervention	9 (20.0)
**Tumor characteristics and pathology**	
Size, mean ± SD (mm)	14.5 ± 16.8
Size ≤ 1 cm	26 (57.8)
Size 1–2 cm	8 (17.8)
Size ≥ 2 cm	11 (24.4)
Endocrine function	0 (0)
Vascular invasion	3 (6.7)
Lymphatic invasion	3 (6.7)
Abnormal chromogranin A (>95 ng/mL)	5 (11.1)
Tumor grade	
Grade 1 (Ki-67 ≤ 3%)	27 (60.0)
Grade 2 (Ki-67 3–20%)	6 (13.3)
Grade 3 (Ki-67 ≥ 20%)	12 (26.7)
Tumor stage	
Stage 1	28 (62.2)
Stage 2	1 (2.2)
Stage 3	5 (11.1)
Stage 4	11 (24.4)
Resection margin R1	3 (6.7)
Resection margin R2	2 (4.4)
**Systemic treatments**	
Somatostatin receptor antagonist	8 (17.8)
Peptide receptor radionuclide therapy	3 (6.7)
Chemotherapy	14 (31.1)
Everolimus/Sunitinib	3 (6.7)
**Outcomes**	
Local recurrence	4 (8.9)
Disease progression	13 (28.9)
De novo metastases	11 (24.4)
Tumor-related mortality	10 (22.2)
Overall mortality	11 (24.4)
Follow-up (months), mean ± SD	46.6 ± 41.0
Overall survival (months), mean ± SD	46.1 ± 41.0
Disease free survival (months), mean ± SD	40 ± 39.4

Values are *n* (%) unless otherwise indicated.

**Table 3 diagnostics-14-01881-t003:** Factors associated with disease progression based on univariable and multivariable analysis. OR = odds ratio, CI = confidence interval, B = logistic regression coefficient; SE = standard error. Bold values are statistically significant.

	Univariable *p*-Value	Multivariable Analysis			
		OR (95% CI)	B	SE	*p*-Value
Tumor grade	**<0.001**	6.422 (0.129–318.540)	1.860	1.992	0.350
Tumor stage	**<0.001**	**7.230 (1.105–47.321)**	**1.978**	**0.959**	**0.039**
Symptomatic presentation	**0.007**	10.929 (0.548–218.038)	2.391	1.527	0.117
Distance from anal verge	0.240	-	-	-	-
Size	**0.041**	0.863 (0.737–1.009)	−0.148	0.080	0.065
Ki-67	**0.004**	1.010 (0.933–1.094)	0.010	0.041	0.798
Basal chromogranin	0.335	-	-	-	-
Positive margins (R1 or R2)	**0.014 ***	-	-	-	-

* Excluded from multivariate analysis for paucity of cases.

## Data Availability

The data presented in this study are available on request from the corresponding author.

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
