# Peer review of "Pathological Characteristics, Management, and Prognosis of Rectal Neuroendocrine Tumors: A Retrospective Study from a Tertiary Hospital"

_diagnostics, 2024, doi:10.3390/diagnostics14171881_

Round 1
Reviewer 1 Report
Comments and Suggestions for Authors
This is a retrospective study from the Neuroendocrine Tumor Registry at a tertiary hospital in Milan, Italy, from 2005 to 2023. Forty-five patients with rectal neuroendocrine tumors were collected and analyzed. The manuscript was well organized and provided a full view of rectal neuroendocrine tumor.
Some suggestions are listed here:
1. The authors provided mean value of Ki-67 index. It will be better to group the patients with Ki-67<3%, 3%-20% and Ki-67>20% (Table 1) according to 2019 WHO classification and grading criteria for neuroendocrine neoplasms of the GI tract. Further univariate and multivariate analysis should be done according to this grouping (Table 3).
2. Please provide the normal range of Chromogranin A level and group the patient according to normal or elevated chromogranin A level (Table 1). Further univariate and multivariate analysis should be done according to this grouping (Table 3).
3. Please provide the proportion of tumor size with tumor <1cm, 1cm-2cm and size>2cm.
4. Figure 1 is confusing. What is the meaning of “no further treatment”. Does it mean that the patient had stable disease without recurrence and no further treatment was needed or the patient died and no further treatment was needed?
5. Please provide a figure legend for each figure.
6. What are the meanings of the abbreviations “PRRT” and “SSA” in Figure 2?
7. Is the size in the univariate and multivariate analysis a continuous variable? Analysis using size groups of <1cm, 1-2cm, and >2cm may be considered.
Author Response
This is a retrospective study from the Neuroendocrine Tumor Registry at a tertiary hospital in Milan, Italy, from 2005 to 2023. Forty-five patients with rectal neuroendocrine tumors were collected and analyzed. The manuscript was well organized and provided a full view of rectal neuroendocrine tumor.
Some suggestions are listed here:
- The authors provided mean value of Ki-67 index. It will be better to group the patients with Ki-67<3%, 3%-20% and Ki-67>20% (Table 1) according to 2019 WHO classification and grading criteria for neuroendocrine neoplasms of the GI tract. Further univariate and multivariate analysis should be done according to this grouping (Table 3).
Thank you for this suggestion, in the text the neoplasms have been analysed according to both the Ki67% index as continuous variable and according to the Grading (G1: Ki67 < 3% or mitotic count <2 in 2 mm2, G2: Ki67 3–20% or mitotic count 2–20 in 2 mm2, and G3: Ki67 > 20% or mitotic count >20 in 2 mm2) as reported in the 2019 WHO classification. This has been clarified in the text in the methods section..
- Please provide the normal range of Chromogranin A level and group the patient according to normal or elevated chromogranin A level (Table 1). Further univariate and multivariate analysis should be done according to this grouping (Table 3).
We have divided the patients according to the Chromogranin A normality levels, as previously described, and changed the table and results accordingly. Regarding the univariate and multivariate analyses in Table 3, we have carefully considered your suggestion, however we have decided to maintain the Chromogranin A as a continuous variable for these analyses. This was done for three reasons: to preserve data granularity (using Chromogranin A as a continuous variable allows us to retain the full spectrum of information, potentially revealing more nuanced relationships between Chromogranin A and other variables), to maintain statistical power, which we would lose by grouping a continuous variable, and to analyze threshold effects.
- Please provide the proportion of tumor size with tumor <1cm, 1cm-2cm and size>2cm.
The proportions of tumor sizes <1cm, 1cm-2cm and >2cm have been reported in table 2.
- Figure 1 is confusing. What is the meaning of “no further treatment”. Does it mean that the patient had stable disease without recurrence and no further treatment was needed or the patient died and no further treatment was needed?
No further treatment indicates both stable disease without recurrence or patient progression requiring only palliative care, as described individually in figure 2. We have added this information in the figure legend.
- Please provide a figure legend for each figure.
Thank you, we have now added figure legends.
- What are the meanings of the abbreviations “PRRT” and “SSA” in Figure 2?
We have now spelled out these abbreviations in the figure legend.
- Is the size in the univariate and multivariate analysis a continuous variable? Analysis using size groups of <1cm, 1-2cm, and >2cm may be considered.
For the reasons stated earlier in the revision, we would prefer not transforming continuous variables in categorical in the univariate and multivariable analysis.

Reviewer 2 Report
Comments and Suggestions for Authors
The authors present a unicenter series of rectal NENs.
As they correctly pointed out, the series is very heterogeneous, but a mix of small cases treated by endoscopy only and stage IV and/or G3 cases is far from allowing any conclusions.
The originality of the manuscript is too limited, the authors should rather focus on a subgroup of rectal NENs and better define the message.
Author Response
The authors present a unicenter series of rectal NENs.
As they correctly pointed out, the series is very heterogeneous, but a mix of small cases treated by endoscopy only and stage IV and/or G3 cases is far from allowing any conclusions.
Thank you, we have further discussed the limitations of the study using your considerations.
The originality of the manuscript is too limited, the authors should rather focus on a subgroup of rectal NENs and better define the message.
As properly suggested, we have expanded the section on endoscopically treated NEN in the results section and in the discussion.

Reviewer 3 Report
Comments and Suggestions for Authors
Thank you very much for a well-written and very interesting manuscript.
This study is of great importance and deserves publication.
I only have a few minor comments.
1. On page 5. The first paragraph consists primarily of data from Table 1. Is it necessary to repeat in the text?
2. There are patients with Ki-67 > 20%. Are they highly or poorly differentiated?
Author Response
Thank you very much for a well-written and very interesting manuscript.
This study is of great importance and deserves publication.
I only have a few minor comments.
- On page 5. The first paragraph consists primarily of data from Table 1. Is it necessary to repeat in the text?
Thank you for this comment. The first paragraph was in fact redundant, and we have removed the repeated information.
- There are patients with Ki-67 > 20%. Are they highly or poorly differentiated?
All but one of these patients had a poorly differentiated neuroendocrine neoplasm. We have now added this in the results section.

Reviewer 4 Report
Comments and Suggestions for Authors
Very interesting paper from all points of view, I liked the corrections, few, but which make it smoother, I liked the summary table of the drugs used and also the summary mirror even the writing is a little small. I would have liked it more if they had been cited in the bibliography PMID: 38051513 for somatostatin and analogues and DOI: 10.1016 /j.trre.2015.09.001 for everolimus. In my opinion, these quotes would complete the essay by framing the problem of neuroendocrine neoplasia from all points of view.
Author Response
Very interesting paper from all points of view, I liked the corrections, few, but which make it smoother, I liked the summary table of the drugs used and also the summary mirror even the writing is a little small. I would have liked it more if they had been cited in the bibliography PMID: 38051513 for somatostatin and analogues and DOI: 10.1016 /j.trre.2015.09.001 for everolimus. In my opinion, these quotes would complete the essay by framing the problem of neuroendocrine neoplasia from all points of view.
Thank you for your kind comments of appreciation. We have now added the references on somatostatin and everolimus.
Round 2
Reviewer 2 Report
Comments and Suggestions for Authors
Although the Authors have modified the study, changes are very few. The originality and scientific soundness of a retrospective, heterogeneous series of rectal NENs is low.
The focus on a subgroup should be the focus of the study (only G3? only advanced? only cases treated by endoscopy?), and not limited to few more sentences added to this version of the manuscript, only describing some details of cases treated by endoscopy.
Author Response
Although the Authors have modified the study, changes are very few. The originality and scientific soundness of a retrospective, heterogeneous series of rectal NENs is low.
The focus on a subgroup should be the focus of the study (only G3? only advanced? only cases treated by endoscopy?), and not limited to few more sentences added to this version of the manuscript, only describing some details of cases treated by endoscopy.
We appreciate the reviewer's concern regarding the heterogeneity of the patient population and the potential impact on the study's originality and scientific soundness. We acknowledge that the inclusion of both early-stage and advanced rectal neuroendocrine tumors (rNENs) introduces a degree of variability. However, we believe that this heterogeneity reflects the real-world clinical scenario and offers valuable insights into the diverse presentations and management approaches for rNENs. The primary objective of our study was to provide a comprehensive overview of rNENs, encompassing their pathological characteristics, management strategies, and prognostic factors. By including a diverse range of cases, we aimed to capture the spectrum of disease presentations and treatment modalities encountered in clinical practice. This approach allows for a more holistic understanding of rNENs and can inform clinical decision-making for a broader patient population.
While focusing on a specific subgroup, such as G3 tumors or advanced-stage disease, may offer a more homogenous dataset, it would limit the generalizability of the findings and potentially exclude valuable information regarding the management of early-stage or less aggressive rNENs. The inclusion of both endoscopic and surgical cases further contributes to the study's comprehensiveness and reflects the current treatment landscape for rNENs. We have carefully considered the reviewer's suggestion to expand the focus on the subgroup of patients treated endoscopically. In response, we have enhanced the Results and Discussion sections to provide a more detailed analysis of this subgroup, including specific outcomes and prognostic factors. We believe that this additional information strengthens the manuscript and addresses the reviewer's concerns regarding the focus on specific subgroups.